# The Biological Impacts of Sitagliptin on the Pancreas of a Rat Model of Type 2 Diabetes Mellitus: Drug Interactions with Metformin

**DOI:** 10.3390/biology9010006

**Published:** 2019-12-25

**Authors:** Lamiaa M. Shawky, Ahmed A. Morsi, Eman El Bana, Safaa Masoud Hanafy

**Affiliations:** 1Department of Histology and Cell Biology, Benha Faculty of Medicine, Benha University, Benha 13511, Egypt; Lamiashawky1974@gmail.com; 2Department of Histology and Cell Biology, Faculty of Medicine, Fayoum University, Fayoum 63511, Egypt; 3Department of Anatomy, Benha Faculty of Medicine, Benha University, Benha 13511, Egypt; emanelbana88@gmail.com; 4Department of Anatomy, Faculty of Medicine for Girls, Al-Azhar University, Cairo 11865, Egypt; Safaahanafy4577@gmail.com

**Keywords:** sitagliptin, immunohistochemistry, insulin, caspase-3, iNOS, HFD/STZ diabetes, pancreas, rat

## Abstract

Sitagliptin, a dipeptidyl peptidase-4 (DPP-4) inhibitor, is a beneficial class of antidiabetic drugs. However, a major debate about the risk of developing pancreatitis is still existing. The aim of the work was to study the histological and immunohistochemical effects of sitagliptin on both endocrine and exocrine pancreases in a rat model of type 2 diabetes mellitus and to correlate these effects with the biochemical findings. Moreover, a possible synergistic effect of sitagliptin, in combination with metformin, was also evaluated. Fifty adult male rats were used and assigned into five equal groups. Group 1 served as control. Group 2 comprised of untreated diabetic rats. Group 3 diabetic rats received sitagliptin. Group 4 diabetic rats received metformin. Group 5 diabetic rats received both combined. Treatments were given for 4 weeks after the induction of diabetes. Blood samples were collected for biochemical assay before the sacrification of rats. Pancreases were removed, weighed, and were processed for histological and immunohistochemical examination. In the untreated diabetic group, the islets appeared shrunken with disturbed architecture and abnormal immunohistochemical reactions for insulin, caspase-3, and inducible nitric oxide synthase (iNOS). The biochemical findings were also disturbed. Morphometrically, there was a significant decrease in the islet size and islet number. Treatment with sitagliptin, metformin, and their combination showed an improvement, with the best response in the combined approach. No evidence of pancreatic injury was identified in the sitagliptin-treated groups. In conclusion, sitagliptin had a cytoprotective effect on beta-cell damage. Furthermore, the data didn’t indicate any detrimental effects of sitagliptin on the exocrine pancreas.

## 1. Introduction

The incidence of diabetes mellitus (DM) is dramatically increasing, particularly in developing countries. In the year 2013, it was reported that about 382 million people had diabetes worldwide, with type 2 diabetes constituting about 90% of the cases. Moreover, the associated morbidity and mortality were raised progressively, making diabetes the 8th leading cause of death worldwide during the year 2012–2013 [1].

Streptozotocin (STZ) is a chemical agent that is often used to induce the experimental models of DM through the destruction of the animal pancreatic beta cells with subsequent hyperglycemia. It can be given in combination with a high-fat diet to establish an animal model of type 2 DM [2].

Metformin is a broadly prescribed oral hypoglycemic medication that can delay the development of type 2 DM or even prevent it, particularly if associated with a modified lifestyle. It acts primarily via increasing insulin sensitization in the liver and peripheral tissues, leading to a decrease in the hepatic glucose output and an increase in peripheral glucose disposal [3].

Currently, the role of incretin hormones is included in the pathogenesis of type 2 DM. Incretins, including glucagon-like peptide-1 (GLP-1) and glucose-dependent insulinotropic polypeptide (GIP), are peptide hormones released by the gastrointestinal tract (GIT) mucosa in response to food intake and stimulate insulin secretion by beta cells, which is so-called incretin effect [4]. These hormones have a short half-life, so they are present in high concentrations only in the postprandial state, being rapidly degraded by dipeptidyl peptidase-4 (DPP-4) enzyme and eliminated in urine [5]. In type 2 DM, an “incretin defect” is manifested through the reduction in incretin bioavailability due to their rapid degradation by dipeptidyl peptidase-4 (DPP-4). Oral administration of dipeptidyl peptidase-4 inhibitors (DPP-4i) has emerged as a new class of antidiabetic agents owing to their ability to extend the biological effects of incretin hormones [6].

Sitagliptin, a DPP-4i, has been shown to significantly control the inflammation, apoptosis, oxidative, and nitrosative stresses in animal models of type I and II DM [7]. Also, the once-daily treatment of type 2 diabetic patients with sitagliptin has revealed a proper glycemic control, as well as reduced glycosylated hemoglobin (HA1c), with no risk of hypoglycemia [8].

Despite its efficacy and long clinical success, sitagliptin and other incretin-based therapies have raised important concerns about the likelihood of developing pancreatitis, on extended use [9]. In recent years, a previous case report study [10] has discussed a patient with no identifiable risk factors for pancreatitis other than sitagliptin use, who presented with pancreatitis based on clinical symptoms and laboratory data. More recently, researchers [11] have evaluated pancreatic safety in the Trial Evaluating Cardiovascular Outcomes with Sitagliptin (TECOS) Study. They reported a small absolute increased risk for pancreatitis with DPP-4i therapy.

Furthermore, post-marketing registration of possible cases of associated acute pancreatitis pushed the FDA to announce a post-marketing safety warning to exenatide, liraglutide, and the dipeptidyl peptidase 4 inhibitor, sitagliptin [12]. Till now, the information on the risk of acute pancreatitis in patients receiving DPP-4i is limited and controversial [13]. Therefore, the current study was established to investigate the biological impacts of sitagliptin on both endocrine and exocrine pancreatic tissues in a rat model of type 2 DM induced by a high caloric diet/low dose streptozotocin (STZ) injection, using correlated biochemical and histological methods. Moreover, a possible synergistic effect of sitagliptin, in combination with metformin, was also evaluated.

## 2. Materials and Methods

### 2.1. Drugs, Chemicals, and Supplements

Streptozotocin (STZ), a chemical powder supplied as 1 g vial, was obtained from Sigma-Aldrich Company (St. Louis, Mo, USA).Sitagliptin, in the form of JANUVIA^®^ 100 tablet was obtained from Merck Sharp and Dohme Ltd. (Pavia, Italy). Each JANUVIA^®^ 100 tablet was ground and dissolved in 10 mL solution of 0.5% carboxymethyl cellulose (CMC), and afterward shaken to obtain a suspension form (10 mg/mL).Metformin, in the form of GLUCOPHAGE^®^ 500 mg tab, was purchased from Minapharm (Cairo, Egypt) under license of Merck Santé France. Each Metformin 500 mg tablet was ground and dissolved in 10 mL solution of 0.5% CMC, and afterward shaken to obtain a suspension form (50 mg/mL).Citric acid, sodium citrate, and sodium carboxymethyl cellulose (Na-CMC) were obtained from ADWIC CO. (Cairo, Egypt). Citric acid and sodium citrate were used for the preparation of the citrate buffer.The diet ingredients such as lard and soybean oil were purchased from commercial sources.

### 2.2. Animals

Fifty adult male albino rats, weighing about 170–210 g, were used in the present study. The animals were locally bred in the animal house of Kasr El-Ainy, Cairo University, Egypt. The rats were housed at an ambient temperature of 26 ± 1 °C, maintained under a natural daily 12-h light/dark cycle. The rats had free access to food and water *ad libitum*. For acclimatization, the rats were handled manually for one week prior to the experiment. All ethical issues regarding animal handling and procedures were followed and approved. All experimental procedures were done in compliance with the Guide for Care and Use of Laboratory Animals published by the US National Institutes of Health [14]. Whenever possible, the procedures in the current study were conducted to avoid or minimize suffering, distress, and pain to animals.

### 2.3. Induction of Type 2 Diabetes Mellitus

To establish a nongenetic rat model of type 2 diabetes mellitus, the fat-fed/STZ rat model was used according to a previous study [15]. The experimental animals were fed with a high-fat diet (HFD) for 3 weeks, then injected by low dose STZ. The high-fat diet (HFD) was composed of 60% fat, 20% protein, and 20% carbohydrates (No. D12492diet; Research Diets, https://researchdiets.com/formulas/d12492). During feeding duration, the weight of the rats was measured every week to ensure the weight gain. Also, fasting blood glucose level was recorded every three days. At the end of the 3 weeks, the rats were fasted overnight, then received a single intraperitoneal injection of freshly prepared STZ (40 mg/kg) in citrate buffer (0.1 mM, pH = 4.5) in a volume of 2 mL/kg. Ten days after STZ injection, the fasting blood glucose level of each rat was recorded in a tail-vein blood sample using an Accu-Chek Active glucometer (Roche Diagnostics, Manheim, Germany). The diabetic rats with fasting blood glucose levels higher than 270 mg/dL were selected according to the model described by Furman [15]. The normal control rats were fed with a basal diet and received a single intraperitoneal injection of the vehicle, citrate buffer. Treatment with sitagliptin and metformin either alone or in combination started after the estimation of blood glucose, on the 10th day after STZ injection (Table 1).

### 2.4. Experimental Design

Fifty rats were used and randomly divided into five groups, ten rats each (Table 1):**Group 1**: Normal control rats were fed a basal diet, received an equivalent volume of citrate buffer solution (parallel to STZ), through the same route of administration.**Group 2:** untreated diabetic rats. The animals of this group were subdivided into 2 subgroups, 5 rats received no treatment (subgroup 2a), and the remaining 5 rats received CMC, orally for 4 weeks (subgroup 2b), parallel to sitagliptin and metformin.**Group 3**: sitagliptin-treated diabetic rats were treated with sitagliptin (10 mg/kg/day) via oral gavage for 4 weeks.**Group 4:** metformin-treated diabetic rats were treated with metformin (200 mg/kg/day) via oral gavage for 4 weeks.**Group 5**: both combined-treated diabetic rats received both sitagliptin (10 mg/kg/day) and metformin (200 mg/kg/day) via oral gavage for 4 weeks.

All animals were fed the corresponding diet throughout the study, a basal diet for control and HFD for the experimental animals.

In accordance with literature, the doses of sitagliptin and metformin (10 mg/kg and 200 mg/kg, respectively) were selected from the low end of the ranges used in previous studies [16,17,18], in order that a potential synergistic effect could be detected and to avoid metformin toxicity if given ≥600 mg/kg/day [19]. The dose of medications was adjusted every week, according to the body weight changes to keep a similar dose per kg body weight of rat, over the whole period of treatment, for each group.

The body weight changes in the experimental groups were measured every 2 weeks to be compared with the control group. Also, the fasting blood glucose was measured every 2 weeks through a tail vein blood sample, using portable Accu-Chek Active glucometer (Roche Diagnostic, Manheim, Germany).

### 2.5. Blood Glucose Tolerance Test (OGTT)

At the end of the treatment period, animals were fasted overnight (12 h) and then orally gavaged a solution of glucose (2.5 g/kg of body weight) according to the literature [17]. Tail vein blood samples were taken every 30 min and for 2 h (0, 30, 60, 90, and 120 min) after glucose administration and checked for glucose level using Accu-Chek Active glucometer.

### 2.6. Biochemical Profile Assays

At the end of the experiment, all animals were anesthetized using ketamine/xylazine (80/10 mg/kg) according to previous work [20], and terminal blood samples were obtained directly from the heart via cardiac puncture. All blood samples were centrifuged, serum was separated and stored at −80 °C for later laboratory analysis of various biochemical parameters. Serum levels of total cholesterol, triglycerides, and lipase were quantified using the corresponding standard procedures and diagnostic kits (Cairo, Egypt), for each, according to the manufacturer’s instructions. Moreover, fasting blood insulin (FBI) was measured using a rat-specific insulin enzyme-linked immunoassay (ELISA) kit (Mercodia AB, Uppsala, Sweden). Fasting blood glucose (FBG) was measured by a glucometer. The insulin resistance (IRI) index was calculated by the homeostasis model of assessment (HOMA-IR) as follows [21]: HOMA-IRI = FBG (mmol/L) × FBI (μU/mL)/22.5.

### 2.7. Histological Procedures

After obtaining the blood samples, the anesthetized rats were euthanized by cervical decapitation. Rats’ pancreases were rapidly dissected, weighed, and immediately fixed in 4% phosphate-buffered formalin for the processing of paraffin sections and histological examination. Nine segments, representative of the different parts of the pancreas, were collected from each pancreas of all the experimental animals. The nine tissue segments of each rat were embedded in 3 paraffin blocks (3 segments per block) so that each block contained a segment from the head, body, and tail of each pancreas. Approximately, five-micrometer sections were cut and were subjected to:Hematoxylin and eosin (H&E) according to previously stated protocols [22].

In H&E stained pancreatic sections, both exocrine and endocrine pancreases were assessed for tissue injury. Acinar damage was evaluated according to the appearance of lining cells, its pyramidal structure, regular patency of the acinar lumen, presence or absence of inflammatory cells, and pyknotic nuclei. Also, ductal changes in the form of epithelial alterations and cellular infiltration were investigated. Damage to the islets of Langerhans was evaluated by the presence or absence of intra-islet hemorrhages, vacuolations, nuclear pyknosis, and cellular infiltrates.

Immunohistochemical techniques, using the peroxidase-labeled streptavidin-biotin method, according to previously demonstrated protocols [23].Immunohistochemical staining was applied for the detection of insulin antibody (INS05 (2D11-H5)). The primary antibody used was mouse monoclonal insulin antibody (Labvision Corporation, Fremont, CA, USA). It was supplied at a dilution of 0.5–1.0 µg/mL and incubated with the slides for one hour at room temperature.Immunohistochemical staining for the detection of the caspase-3 antibody as a marker of apoptosis. The primary antibody used was rabbit polyclonal caspase-3 antibody (Labvision Corporation, Fremont, CA, USA). It was supplied at a dilution of 1:100 and incubated with the slides for one hour at room temperature.Immunohistochemical staining for the detection of inducible nitric oxide synthase (iNOS) antibody as a marker of inflammation. The primary antibody used was rabbit polyclonal iNOS antibody (Labvision Corporation, Fremont, CA, USA). It was supplied at a dilution of 1 µg/mL and incubated with the slides for one hour at room temperature.

The sections were incubated with the corresponding primary antibody diluted to the corresponding concentration in PBS for one hour, followed by a reaction with a biotinylated secondary antibody. After conjugation with streptavidin-biotin-peroxidase complex, 3,3-diaminobenzidine (DAB) was used as a chromogen, and hematoxylin solution was used as a counterstain. The reaction gave brownish discoloration in the cytoplasm of the pancreatic cells. All tissue specimens were examined by light microscopy. Image acquisition was performed with a digital microscope camera (Leica Qwin 500, Leica, Cambridge, England) computer system.

### 2.8. Histomorphometric Analysis

The data were obtained using the image analyzer computer system (Leica Qwin 500, Leica, Cambridge, England). Three histological sections per slide with a total of nine per animal were used in the morphometric study. The examination was done in non-serial H&E and immunostained pancreatic sections. The image analyzer was first calibrated automatically to convert the measurement units (pixels) produced by the image analyzer program into actual micrometer units. It was used for:Assessment of the islet number and islet size: For each study group, the islets were counted, and islet size was evaluated in 10 different, randomly selected microscopic fields, using lower magnification. Then, the mean islet number per field and mean islet size were calculated for each group of animals. The number of islets was expressed as N/10 mm^2^ of the pancreatic parenchyma, according to Noor et al. [24]. For the islet size, the maximum diameter of the islet was selected by comparing all possible radii diameters per islet and choosing the greatest according to previous work [4].Evaluation of the area % of insulin, caspase-3, and iNOS immunostaining: They were measured in 10 non-overlapping high power fields, using the interactive measurements menu. The brown coloration of the immunoreaction was covered automatically by a blue mask (binary image). The area of this binary image was then calculated, which reflected the positively stained cells for insulin, caspase-3, and iNOS.

### 2.9. Statistical Analyses

The values obtained were expressed as mean ± SD for each group. Statistical comparison among different groups was evaluated using a one-way analysis of variance (ANOVA) and post hoc Least Significant Difference (LSD) test. Calculations were done with SPSS software, version 20 (IBM SPSS Statistics for Windows, Armonk, NY, USA). Statistical significance was defined as *p*-value less than 0.05.

## 3. Results

### 3.1. Mortality, Food Intake, Water Intake, and Changes in Body Weight, Pancreas Weight, and Pancreas Weight to Body Weight Ratio

No mortality was observed among the rats of different groups. There was no significant change in food consumption or water intake among the different treated groups when compared to the untreated diabetic group. As shown in Table 2, on week 0 (after induction of diabetes and before starting the treatments), all diabetic rats showed a significant increase in body weight compared to the control group (*p <* 0.05). At the end of the treatment period (week 4), weight gain was significant in the untreated diabetic animals (group 2) when compared to the control one (*p <* 0.05). In both sitagliptin- and metformin-treated rats (groups 3 and 4, respectively), the body weight was significantly maintained (*p <* 0.05), compared with the untreated diabetic group with better preservation of the body weight in combined therapy group (*p <* 0.05). In regards to the pancreatic weight and pancreas/body weight ratio, both parameters significantly reduced (*p <* 0.05) in untreated diabetic rats at the end of 4 weeks, when compared to control. Sitagliptin therapy alone significantly maintained pancreatic weight and pancreas/body weight ratio compared to the untreated diabetic group (*p <* 0.05), but still less than control. In metformin-treated animals, both parameters were slightly but significantly maintained (*p <* 0.05), compared to the untreated diabetic rats. However, the combination of both agents showed better preservation of pancreatic weight and pancreas weight/body weight ratio than monotherapy groups.

### 3.2. Glucose Homeostasis Parameters

After the induction of diabetes and prior to initiation of therapy (week 0), all diabetic rats showed a significant increase (*p <* 0.05) in the level of FBG compared to control animals (Figure 1). All treated diabetic rats showed a variable reduction in the FBG during the treatment period. During this treatment period, it was apparent that all treated diabetic rats had significantly (*p <* 0.05) lower FBG compared to untreated diabetic ones. Whereas the combination of sitagliptin and metformin had a better synergistic effect on glucose control when compared to either agent alone (*p <* 0.05) (Figure 1).

Before the initiation of treatment, all diabetic rats showed a significant decrease (*p <* 0.05) in serum insulin, compared to their control (Table 3). By the end of the treatment period, the acquired results (Figure 2) revealed a significant decrease in serum insulin levels in untreated diabetic animals when compared to the control group (*p <* 0.05). The sitagliptin-treated group showed a significant increase in serum insulin level when compared to the untreated diabetic one (*p <* 0.05), while a non-significant change was noted in those metformin-treated. Furthermore, sitagliptin treatment revealed significant improvement in comparison to metformin treatment (*p <* 0.05). Actually, the combined treatment in group 5 showed a significant increase (*p <* 0.05) in the serum insulin level, with almost normalization of the value, when compared to the control group.

The insulin resistance index was calculated by the HOMA-IR equation using the level of fasting insulin (µIU/mL) and fasting glucose (mmol/L). The baseline values of HOMA-IR (before initiation of therapy) showed a significant increase (*p <* 0.05) in all diabetic rats compared to control (Table 3). By the end of the treatment period, all treated diabetic animals showed a significant decrease in HOMA-IR, when compared to the untreated diabetic group (*p <* 0.05) with almost correction of the insulin resistance index, in the combined therapy group (Figure 2). No statistical significance was noticed between sitagliptin and metformin-treated groups, regarding HOMA-IR values.

In regards to the OGTT conducted at the end of the treatment period, the acquired result of the test (Figure 3) revealed an impairment in the fasting glucose tolerance and impaired glucose tolerance during the 2 h of the test in untreated diabetic rats, compared to control (*p <* 0.05), which was demonstrated by a greater area under the curve (AUC), in this group. However, all treated groups showed reduced AUC compared to an untreated diabetic group, indicating no impairment of the fasting glucose tolerance but impairment (*p <* 0.05) after glucose administration to a lesser extent than in the untreated group. Moreover, the AUC was much smaller (*p <* 0.05) in the combined therapy group than in monotherapy groups, which indicated the best response in the OGTT.

### 3.3. Lipid Profile

The baseline values of total cholesterol and triglycerides (TGs), before initiation of treatment, showed a significant increase (*p <* 0.05) in both parameters in the untreated diabetic rats compared to the control animals (Table 4). At the end of the treatment period, sitagliptin, metformin, and both combined significantly corrected the level of total cholesterol and TGs in all treated groups when compared to the untreated one (*p <* 0.05). However, almost complete normalization of the values of both parameters was observed in the combined therapy group. Noteworthy, sitagliptin alone treatment showed slight but significant improvement (*p <* 0.05) in cholesterol levels only in comparison to metformin-treated animals (Table 5).

### 3.4. Serum Lipase

The results of laboratory assay of lipase activity, before and after treatment, revealed a non-significant difference in the serum level of lipase in all treated and untreated diabetic rats when compared to control animals (Table 5).

### 3.5. Islet Cells Morphology and Pancreatic Acini

H&E stained pancreatic sections of the control animals showed regular and well-defined islets of Langerhans. They appeared pale staining areas, surrounded by deeply stained peripherally located pancreatic acini. The islets cells had a cord-like cellular arrangement with intervening blood capillaries. The endocrine cells of the islets appeared with pale acidophilic cytoplasm and central vesicular nuclei. The exocrine pancreas showed normal histological structure in the form of regularly, closely packed, patent acini lined by pyramidal cells with apical acidophilia and basal basophila (Figure 4A).

Regarding the combination of 3 weeks’ HFD with low dose STZ injection in group 2, all specimens (obtained from the diabetic animals receiving CMC and those not receiving) showed markedly depleted islets of Langerhans. A concomitant decrease in both the size of islets as well as the number of cells per islet was evident. The islets appeared shrunken, hypocellular with nuclear pyknosis (Figure 4B). Other sections showed disrupted histoarchitecture (loss of the normal cellular cord arrangement), with nuclear pyknosis, vacuolar changes, and congested blood capillaries (Figure 4C). No obvious histological changes were seen in the surrounding pancreatic acini of the different sections of all specimens. They appeared closely packed with no cellular infiltration, hemorrhage, or degenerative changes (Figure 4B).

Treatment with sitagliptin (Figure 4D), metformin (Figure 4E), and their combination (Figure 4F) showed a marked improvement in islets’ size and histoarchitecture with still minimal apoptotic changes, which may be slightly marked in metformin-treated group, in the form of small darkly stained pyknotic nuclei and deeply stained acidophilic cytoplasm (Figure 4E). The pancreatic acini appeared almost with normal microstructure in all specimens obtained from sitagliptin-alone or combined treatment groups. No areas of vascular congestion, hemorrhage, necrosis, fibrosis, or inflammatory infiltrates were seen in the interstitial tissues (Figure 4D,F). No prominent ductal changes were observed.

### 3.6. Immunohistochemical Observations

Regarding the immunohistochemical localization of insulin-secreting islet beta cells, the islet cells of the control rats displayed a large markedly positive β-cell core surrounded by a mantle zone of negative non-β-cells (Figure 5A). In the untreated diabetic group, shrunken islet showed a reduction in insulin immunoreactivity, and only some β cells displayed minimal insulin immunoreaction (Figure 5B). Treatment of the animals with sitagliptin (group 3), metformin (group 4), or both combined (group 5) showed variable degrees of maintaining the insulin immunoreaction, which was marked in combined therapy group (Figure 5C–E, respectively).

Caspase-3 immunostained sections revealed negative immunoreaction in the islet cells of the control rats (Figure 6A). However, in the untreated diabetic group, shrunken pancreatic islets showed remarkable positive cytoplasmic immunoreactivity widely distributed among islet cells (Figure 6B). Treatment with sitagliptin (Figure 6C), metformin (Figure 6D), or both combined (Figure 6E) showed variable degrees of much less positive immunoreactivity, with marked improvement in the combined therapy group.

Immunohistochemical observations of iNOS stained pancreatic sections revealed negative immunoreaction in the islet cells, as well as the pancreatic acini of the control rats (Figure 7A). However, in the untreated diabetic group (Figure 7B), pancreatic islets showed remarkable positive iNOS cytoplasmic immunoreactivity. The pancreatic acini showed negative immunoreaction. All treated groups showed negative iNOS immunoreactivity in both endocrine and exocrine pancreases. Sitagliptin alone or combined showed no effect on the pancreatic acini (Figure 7C–E).

### 3.7. Histomorphometric Results

#### 3.7.1. The Mean Islet Number and Islet Size

Histomorphometric evaluation of the islet number and islet size (Figure 8) revealed a significant decrease in both parameters in the untreated diabetic rats compared to the control animals (*p <* 0.05). However, these morphometric parameters were significantly increased (*p <* 0.05) in all treated diabetic groups when compared to the untreated diabetic group. Moreover, they showed a highly significant increase in the combined therapy group when compared to monotherapy groups (*p <* 0.05). Meanwhile, sitagliptin-treated animals showed a significant increase (*p <* 0.05) in both the number and size of pancreatic islets compared to those metformin-treated.

#### 3.7.2. Area Percent of Insulin Immunoreactivity

Figure 9A shows a significant decrease (*p <* 0.05) in the mean area % of insulin immunostaining in the untreated diabetic group when compared to control animals. However, in all treated diabetic animals, a significant increase (*p <* 0.05) in the measured parameter was observed when compared to the untreated diabetic group. Sitagliptin alone significantly improved (*p <* 0.05) insulin immunoexpression in B cells, more than metformin. Moreover, the combination with metformin showed a highly significant increase (*p <* 0.05) in the area % of insulin, compared to either agent alone.

#### 3.7.3. Area Percent of Caspase-3 Immunoreactivity

The statistical data, shown in Figure 9B, revealed a significant increase (*p <* 0.05) in the mean area % of caspase-3 immunostaining in the untreated diabetic group (group 2) when compared to control animals (group 1). All treated diabetic rats showed a significant decrease (*p <* 0.05) in the caspase-3 immunostaining compared to those untreated. However, the independent use of each drug showed a significant decrease (*p <* 0.05) in sitagliptin-treated animals more than those metformin-treated. In the combined therapy group, there was a significant decrease (*p <* 0.05) in the caspase-3 immunostaining when compared to either sitagliptin or metformin-treated groups.

#### 3.7.4. Area Percent of iNOS Immunoreactivity

Figure 9C shows a significant increase in the area % of iNOS immunoreactivity in the untreated diabetic group when compared to the control animals (*p <* 0.05). All treated diabetic groups showed a significant decrease in the iNOS immunoreactivity when compared to the untreated one (*p <* 0.05). No difference in statistical significance was present between the treated diabetic groups.

## 4. Discussion

Sitagliptin, a selective dipeptidyl peptidase-4 (DDP-4) inhibitor, is a relatively new antidiabetic drug, which inhibits the enzymatic activity of dipeptidyl peptidase-4 with subsequent blockage of the rapid degradation of the glucagon-like peptide-1 (GLP-1) [25]. GLP-1 stimulates both the synthesis and release of insulin from beta cells and reduces the release of glucagon from alpha cells. Therefore, DDP-4 inhibitors exert their glucoregulatory effect through prolonging the duration of the active circulating incretin hormones in the bloodstream [10].

Interestingly, a key challenge in the long term use of sitagliptin, in chronic management of type 2 DM, is its possible adverse effects on the exocrine pancreas, including acute pancreatitis, as reported in the literature [26,27]. Due to the seriousness and clinical significance of pancreatitis, the current study was designed to evaluate the acute biological effects of 4-weeks sitagliptin treatment on both exocrine and endocrine pancreas and to investigate a potential synergistic action of sitagliptin, when combined with metformin.

In humans, most insulin-resistant obese individuals show an increase in their insulin secretion and remain nondiabetic [28]. Later on, the combination of further beta-cell dysfunction precipitates the development of diabetes due to the failure of the beta cells to compensate for more insulin release during insulin resistance [29]. Given that both insulin resistance and impaired β cell function constitute the hallmarks of the pathogenesis of human type 2 DM [30], high-fat diet-fed, streptozotocin-treated (HFD/STZ) rat model was used to establish a nongenetic rat model of type 2 DM in order to further mimic the pathology of such human disease. While HFD induces insulin resistance, the low dose of STZ causes mild impairment in β cell function [31].

The results presented herein provided compelling evidence regarding the biological effects of sitagliptin on the pancreatic cells and favor its use, in combination with metformin, in controlling HFD/STZ-induced diabetes in rats.

Foremost, there were no great differences in the biochemical or histological findings obtained from the untreated diabetic rats (subgroup 2b) receiving oral CMC (vehicle of the drugs) and those not received (subgroup 2a), so both subgroups were considered the same.

In the current study, the untreated diabetic animals showed a significant weight gain at the end of the experiment, compared to their control littermates. However, the individual use of either sitagliptin or metformin alone, in monotherapy groups, equally maintained body weight (prevent about 51% increase in the body weight) compared to the untreated diabetic rats, although both combined significantly preserved body weight (prevent about 71% increase in the body weight). This means that both agents combined had a synergistic effect on restraining the increase in body weight. Similarly, Matveyenko et al. [26] reported a synergistic effect of the combination of both sitagliptin and metformin in controlling body weight in a human islet amyloid polypeptide (HIP) transgenic rat model of type 2 diabetes, rather either agent alone. In addition, Reimer et al. [32] reported similar findings in the Zucker diabetic fatty rat model of type 2 DM. These results indicated an effective combination of both sitagliptin and metformin in maintaining body weight in different animal models of type 2 DM.

Differently, in the same HFD/STZ rat model of T2DM, Saad et al. [17] revealed weight neutral effects of the individual use of sitagliptin and metformin compared to the control animals. This difference might be attributed to the method of induction of DM, in particular, the duration of HFD and the dose of STZ.

In regards to the pancreatic weight and pancreas/body weight ratio, both parameters were significantly reduced in untreated diabetic animals compared to their control rats, in accordance with previous works [33,34]. Oral administration of sitagliptin alone significantly increased both parameters when compared to the untreated diabetic rats. However, metformin slightly but significantly increased pancreatic weight and pancreas/body weight ratio when compared to the same untreated diabetic animals. Noteworthy, the combined therapy group showed a significant increase in the pancreatic weight and pancreas/body weight ratio in comparison to either of the monotherapy groups.

The main goal of antidiabetic drugs is to decrease or maintain blood glucose levels as tightly as possible and hence to minimize the development of complications. Although, the response to these drugs is individualized and can show a great difference, most probably due to the multifactorial nature of the pathophysiology of T2DM [35]. In the current experiment, the HFD/STZ rat model showed disruption of glucose homeostasis in untreated diabetic animals. The untreated rats showed a significant increase in blood glucose levels when compared to the control group, in addition to an impairment of the OGTT at the end of the experiment. Oral administration of sitagliptin (in group 3), metformin (group 4), and both combined (group 5), at the mentioned dose, significantly reduced blood glucose levels during the treatment period and showed an improvement in the glycemic control in diabetic rats, as further evidenced by an amelioration of the OGTT. However, the combination of both drugs showed better control of FBG than either agent alone. Given that oral metformin could inhibit DPP-4 activity in T2DM patients, as elucidated by Lindsay et al. [36], the further anti-DPP4 activity could be afforded when combined with sitagliptin. This notion could explain the synergistic effect of sitagliptin on glucose homeostasis when combined with metformin. Also, an additive effect of such combination was mentioned and explained by Andersen et al. [25]. Metformin appeared to stimulate the endogenous GLP-1 secretion, and protection of the elevated levels of incretin hormones was activated by sitagliptin.

Parallel to the changes in blood glucose level in the different studied groups, serum insulin showed significant reduction in untreated diabetic group compared to control rats, in accordance with [37,38,39], who reported a reduced serum insulin level in HFD/STZ rat model of type 2DM, and in controversy with the study of Saad et al. [40] in which the diabetic rats exhibited significantly high serum insulin in the same rat model. The suggested explanation for this controversy may be the duration of HFD feeding in the aforementioned study. It was 4 weeks, unlike the current study, which could exacerbate more insulin resistance in diabetic rats. Therefore, insulin failed to act adequately on resistant tissues with subsequent poor glucose utilization, so β-cells compensated initially for insulin resistance by increasing insulin secretion.

All treated diabetic animals showed different responses regarding the serum insulin level in comparison to the untreated one. The sitagliptin-treated group showed a significant increase in serum insulin, while a non-significant change was noted in those metformin-treated compared to untreated diabetic animals. Furthermore, sitagliptin treatment revealed a significant improvement in comparison to metformin treatment. Actually, the combined treatment in group 5 showed a significant increase in the serum insulin level, with almost normalization of the value, when compared to the control group. In agreement, Matveyenko et al. [26] revealed that a more synergistic effect, on the glucose-mediated insulin secretion, was obtained when both sitagliptin and metformin were combined in a HIP transgenic rat model of type 2 DM. Also, Eitah et al. [41] reported a significant increase and normalization of serum C-peptide level, when they utilized sitagliptin, in combination with quercetin, in STZ-induced diabetic rats. This indicated that sitagliptin, in combination with another antidiabetic agent, further improved insulin release in different models of experimental DM.

In regards to the insulin resistance index (HOMA-IR), the untreated diabetic group showed a significant increase in the IR when compared to the control animals. Similarly, previous studies [17,42] reported significantly higher HOMA-IR values in untreated diabetic rats compared to the control animals.

However, all treated diabetic rats in different studied groups showed a significant decrease in insulin resistance compared to the animals of the untreated diabetic group. Likewise, Saad et al. [17] reported similar results with the least value observed in metformin-treated rats compared to those sitagliptin-treated, which was inconsistent with our study. The current result showed no significant difference between sitagliptin and metformin-treated animals regarding IR. The result presented herein strongly suggested an insulin resistance decreasing effect of sitagliptin, which might be equal to that of metformin. Consistent with this notion, Zheng et al. [43] reported an ameliorative effect of sitagliptin on the development of insulin resistance in ob/ob mice. Furthermore, no significant difference was present in the combined treatment group when compared to the control group. This indicated a complementary action of the two drugs, on decreasing the IR in the combined therapy group.

Taken all together, the determined parameters of glucose homeostasis could be correlated. All treated diabetic animals showed a variable statistical difference in FBG, insulin level, and HOMA-IR in comparison to the untreated diabetic group. The significant decrease in FBG in sitagliptin and the metformin-treated group was accompanied by a significant increase in the serum insulin level in the sitagliptin-treated group but not in metformin-treated one. On the other hand, it was associated with a significant decrease in IR in both groups. Notably, the combination therapy group showed a more potent synergistic effect on these parameters when compared to the same untreated diabetic animals. This indicated that the anti-hyperglycemic effect of sitagliptin, in the current rat model of HFD/STZ-induced diabetes, was mostly afforded through an insulin secretagogue effect and increasing peripheral glucose utilization. On the other hand, the different glucose-lowering mechanisms encountered in each monotherapy group can be summarized to synergistically promote further insulin secretion by beta cells and enhanced insulin sensitivity of the peripheral tissues in the group of combined treatment modality.

Moreover, the disrupted glucose homeostasis in group 1 was associated with abnormalities in the lipid profile as manifested by a significant increase in both total cholesterol and TGs, in accordance with previous publications [44,45]. Also, the increased lipid profile comes hand by hand with the increase in the body weight of the untreated diabetic rats. These abnormalities may be a cause or a consequence of the diabetic state and insulin resistance.

However, the independent and concomitant treatments of sitagliptin and metformin remarkably improved lipid profile compared to the untreated diabetic rats. Likewise, previous researches [17,46] reported that both sitagliptin and metformin could normalize FBG and significantly corrected the level of cholesterol and TGs. Furthermore, sitagliptin alone treatment significantly decreased the level of cholesterol, compared to the untreated diabetic group. However, the combination of both sitagliptin and metformin completely normalized the cholesterol level in the combined therapy group. The improved lipid profile in treated diabetic groups might be correlated with the restraining of the increase in body weight.

Compared to the metformin-treated group, sitagliptin treatment either alone or combined with metformin showed a significant decrease in the cholesterol level. This observation indicated that sitagliptin treatment had a better cholesterol-lowering effect more than metformin, and the combination of both could normalize the cholesterol values. Consistent with an early study [17], comparing metformin, glimepiride, and sitagliptin, it was found that sitagliptin treatment completely normalized the level of cholesterol. Inconsistent with the same study [17], our result showed no significant difference between sitagliptin and metformin-treated rats in TGs lowering effect. This inconsistency between the results could be explained by a difference in the constituents of the HFD, the dose of STZ, duration of feeding, and other different methodologies.

Similar lipid-lowering effects have also been observed with another DPP-4 inhibitor in ZDF rats [12] and STZ-induced diabetic rats [47]. It is still questionable whether the lipid-lowering effect of sitagliptin in animal studies is due to enhanced activity of the incretin hormone, improvement of the glycemic status, or due to delaying gastric emptying time [48,49,50].

Apart from the potent effect of sitagliptin alone on total cholesterol level, the animals of the combined therapy group showed approximate normalization of cholesterol and TGs values when compared to control animals. Overall, the result presented herein strongly suggested that sitagliptin, in combination with another antidiabetic agent, produced more synergistic effects on cholesterol and TGs, as evidenced in previous studies [18,41,51], which reported a more potent effect of sitagliptin when co-administrated with glimepiride, trigonelline, and quercetin, respectively. Owing to the heterogeneous nature of the pathophysiology of T2DM, sitagliptin/metformin combination provides an attractive initial combination therapy due to their complementary mechanisms of action, targeting more than one pathophysiological defect in type 2 diabetes [52].

Serum lipase is an important pancreatic enzyme, mainly produced by acinar cells. Estimation of serum lipase is now more preferred than amylase as a biomarker for pancreatitis due to its improved sensitivity, particularly in alcohol-induced pancreatitis. Its persistent elevation of up to 2 weeks creates a wider diagnostic window than amylase [53]. Induction of diabetes in the diabetic group showed no significant difference when compared to control. At the end of the experiment, all treated and untreated diabetic rats showed no significant change when compared to the control animals. The result presented herein was in accordance with other incretin-based therapy studies. Tatarkiewicz et al. [54] and Usborne et al. [55] reported that animal treatment with GLP-1 receptor agonists—exenatide and dulaglutide—respectively, did not modify plasma level of lipase in different models of DM.

In the current study, all specimens obtained from the untreated diabetic animals (group 2) showed abnormally structured pancreatic islets. They appeared shrunken, hypocellular with altered histoarchitecture in H&E stained sections. In addition, the islet cells showed vacuolated cytoplasm, small pyknotic nuclei, and congested blood capillaries. These findings were consistent with other studies [33,39], which reported that HFD, in combination with low dose STZ injection, produced marked distortion of the islet histology, apoptotic changes, significant depletion of the islet number and islet size, as well as the number of cells per islet. Different mechanisms have been postulated to explain islet cell injury, particularly β-cell damage in type 2 DM. These include increased oxidative stress, accelerated metabolic stress, increased endoplasmic reticulum stress, activation of inflammatory pathways, and toxic accumulation of islet amyloid polypeptide, and the end result is β cell de-differentiation and apoptosis [56].

It has been well known that hyperglycemia and hyperlipidemia are the fuel of certain biochemical pathways, such as oxidative stress, low-grade inflammation, and apoptosis. Actually, these biochemical alterations precipitate the development of insulin resistance and beta-cell dysfunction [57].

The histological changes come hand by hand with the morphometric results, which showed a significant reduction in the islet size and islet number compared to the control animals. Furthermore, these observations were parallel to the biochemical findings of a significant decrease in the serum insulin level and a significant increase in blood glucose in comparison to the control group. The insulin deficiency, in association with increased insulin resistance, as indicated by a significant increase in HOMA-IR, resulted in an elevation of the blood glucose level. Similarly, these findings of disturbed parameters of glucose homeostasis were in accordance with Hussien et al. [39], who reported insulin deficiency, hyperglycemia, and increased HOMA-IR in the same rat model of type DM.

Notably, when correlating the islet number and islet size with the pancreatic weight, both depleted parameters could be an explanation for the significant reduction in the pancreatic weight of the animals of the untreated diabetic group due to affection of the islet mass, as demonstrated by Kou et al. [58]. They suggested that islet number rather than islet size is a major determinant of β- and α-cell mass in humans.

Compared to the untreated diabetic group, treatment with sitagliptin, metformin, and their combination in the different studied groups 3, 4, and 5, respectively, showed an improvement in the islet cell morphology with still minimal apoptotic changes in the form of small-sized darkly stained pyknotic nuclei and deep acidophilia of the cytoplasm of few endocrine cells of the islets. Taken into account the morphometric results of the treated groups, there was a significant increase in the islet size and islet number in all treated groups when compared to the untreated diabetic one. Sitagliptin alone was superior to metformin alone in maintaining islet size and islet number. However, their combination was the best to maintain the measured parameters when compared to the control group.

These findings were consistent with the results of Mu et al. [48], who revealed that sitagliptin preserved beta-cell mass and morphology in the HFD/STZ mice model of DM. In addition, Mohamed et al. [18] reported a better effect of sitagliptin on the islet cell histoarchitecture and islet diameter than glimepiride. However, the combination of both sitagliptin and glimepiride revealed a potent effect than the individual use of these drugs. Moreover, in a HIP rat model of type 2 DM, Matveyenko et al. [26] reported that sitagliptin preserved beta-cell mass, no effect of metformin, while a synergistic effect of their combination on beta-cell mass and morphology. Meanwhile, in a rat model of STZ-induced diabetes mellitus, Eitah et al. [41] reported approximate similar histological effects of the individual use of sitagliptin and quercetin on the islet cell morphology and islet number. Also, they reported that sitagliptin, combined with quercetin, maintained the histology and normalized the islet number. Overall, these observations indicated a better synergistic effect of sitagliptin on the islet cell morphology and morphometry, when combined with another antidiabetic agent, in different animal models of DM. Similarly, the improved morphometric measurements of the islet number and islet size in all treated groups come parallel to the changes in pancreatic weight and could be an explanation for the increased pancreatic weight in the corresponding treated diabetic groups.

When correlating the histological changes in the islet cells morphology with the biochemical findings of all treated groups, the observed increased insulin level and decreased blood glucose concentration may be due to the amelioration of the histological structure of beta cells together with decreased insulin resistance as indicated by a significant decrease in HOMA-IR. The increased insulin release by the preserved beta cells, in addition to improved insulin sensitivity, enhanced glucose utilization in skeletal muscle and peripheral tissues with a subsequent decrease in blood glucose levels.

Regarding the exocrine pancreas, there was a normal structural arrangement of the pancreatic acini with no evidence denoting pancreatitis, such as hemorrhage, necrosis, inflammatory cell infiltrates in all sections obtained from all animals of the treated diabetic groups and, in particular, sitagliptin alone-treatment and sitagliptin plus metformin-treatment groups. This indicated that sitagliptin didn’t induce pancreatitis in this animal model of HFD/STZ type 2 DM. The H&E results of the pancreatic acini were supported by the biochemical finding of serum lipase that revealed non-significant changes in sitagliptin-treated groups. These results were parallel to that of Mega et al. [4], who revealed that sitagliptin treatment prevented the aggravation of inflammation and fibrosis elicited in both exocrine and endocrine pancreases in the ZDF rat model of type 2 DM. Similarly, in incretin-based therapy, Tatarkiewicz et al. [54] showed that exenatide did not evoke pancreatitis in HFD/STZ rat model and could attenuate chemically-induced pancreatitis in normal and diabetic rodents.

In clinical studies, similar findings were in accordance with the current result. A population-based cohort study revealed that DDP-4 inhibitors were less likely to cause drug-induced pancreatitis than sulfonylurea. Also, in a randomized controlled trial, Smits et al. [59] reported no development of pancreatitis in type 2 diabetic patients receiving sitagliptin, even though there was a brief and modest increase in plasma lipase and amylase.

Inconsistent with current results, Nachnani et al. [9] reported pancreatic inflammation, pyknotic nuclei, and cellular infiltrates in normal rats treated with exenatide. In addition, the aforementioned study [26] that revealed a beneficial effect on islet cells reported an additional adverse effect of sitagliptin on the exocrine pancreas in the form of increased pancreatic duct turnover, ductal metaplasia, and, in one rat, pancreatitis. An interesting finding was revealed by the authors that metformin combination prevented sitagliptin-induced ductal proliferation and pancreatitis. Furthermore, in a more recent study, Rouse et al. [27] reported that both sitagliptin and exenatide could induce pancreatic acinar cell injury in mice. They added that HFD fed mice were more exposed than those on a standard diet.

An explanation was postulated to elucidate the possible underlying mechanisms of such drug-induced pancreatic changes: the glucagon-like peptide 1(GLP-1) receptor is normally expressed in the pancreatic duct and islet’s cells and is indirectly stimulated by DPP-4 inhibitors as they lead to a further increase of the circulating level of GLP-1 [60]. Prolonged GLP-1 stimulation resulted in the expansion of pancreatic acinar and ductal cells and their growth into the small pancreatic ducts; this could lead to ductal occlusion, thereby triggering pancreatitis [61]. An intriguing observation is that the above-mentioned studies [26,27] used very high doses of DPP-4 inhibitor (sitagliptin, 200 mg/kg), which are not used in humans. So, their results are inapplicable and not reproducible.

Regarding the immunohistochemical observations, the mean area % of insulin immunoreactivity showed a significant decrease in the untreated diabetic group when compared to the control animals. This result was similar to that of a previous work [62], which reported a decreased insulin localization in the untreated diabetic group. Being insulin secretion highlighting the beta-cell function, previous studies [63] suggested that loss of beta-cell mass and beta-cell function were encountered in the pathogenesis of type 2 DM.

Sitagliptin-treated and metformin-treated groups showed a significant increase in comparison to the untreated diabetic one with a better response in sitagliptin-treated rats. However, the combination of both showed a more potent insulin immunoexpression. The current findings were parallel to a previous study [41], which showed improved insulin immunoreactivity in the sitagliptin-treated diabetic group; however, the combination with quercetin restored the intensity of insulin immunostaining. These results point to a preservative effect of sitagliptin on islet cell function, more than metformin with a potent synergistic action when both combined.

Caspases are protease enzymes that require the presence of cysteine to perform their catalytic activity. Their proteolytic effect lies specifically in an aspartate residue, and hence they are so-called caspase (Cysteine-dependent ASPartate-specific peptidASE). The caspase-cascade system is entangled in the induction, transduction, and amplification of intracellular apoptotic signals. Caspase-3 is one of the executer caspases involved in apoptosis [64]. Caspase-3 immunostained sections obtained from the untreated diabetic group showed a significant increase in the cytoplasmic immunoreactivity when compared to the control group. Similarly, Liu et al. [65] reported the upregulation of caspase-3 expression in STZ-induced DM. It has been well known that the upregulation of the apoptotic cascades in islet cells is involved in beta-cell damage and dysfunction [66].

All treated diabetic groups showed a significant decrease in caspase-3 immunoreaction when compared to the untreated diabetic one, with the least reaction in the combined therapy group. These results indicated variable anti-apoptotic effects of sitagliptin and metformin. In agreement, Samaha et al. [67] reported a significant suppression of caspase-3 immunoexpression in the sitagliptin-treated group. Given that apoptosis is entangled in the pathogenesis of beta-cell dysfunction, and since sitagliptin and metformin altered the caspase-3 expression with the best response in a combined approach, these data confirm the potential role of both agents in alleviating apoptotic changes with an additive effect, when both combined.

Inducible nitric oxide synthase (iNOS) is one of the three isoforms of NOS enzymes involved in the production of nitric oxide. iNOS is not detectable during normal conditions; however, it is expressed in the tissues during inflammation [68]. iNOS immunostained sections in the untreated diabetic group showed a significant increase in the cytoplasmic immunoreactivity of the pancreatic islets when compared to the control group. Similarly, Sai et al. [69] detected positive iNOS immunostaining in the pancreatic islets of the STZ-induced diabetic rat model of DM. It is broadly accepted that the induction of iNOS in pancreatic islets leads to an increase in NO production associated with dysfunctional beta-cells [70].

All treated diabetic groups showed a negative iNOS immunoreaction in the pancreatic islets as well as pancreatic acini. These observations indicated the alleviation of the inflammatory state of the pancreatic islets with treatment. In addition, no inflammation was elicited by sitagliptin in the exocrine pancreatic tissue of groups 3 and 5. In agreement, Salehi et al. [71] reported the expression of iNOS in the pancreatic islets of type 2 diabetic Goto-Kakizaki (GK) rats and counteracted by GLP-1 administration. Considering inflammatory cytokines and NO as potential mediators of pancreatic beta-cell destruction in diabetes, and since sitagliptin and metformin suppressed iNOS expression in the islet cells, these data point out to the same direction as ours and emphasize the potential benefit of both agents in such condition.

Taken all together, daily oral administration of sitagliptin in group 3 maintained body weight, decreased serum glucose and insulin resistance with concomitant enhancement of insulin secretion, decreased total cholesterol and TGs, and increased islet number and islet size. The ameliorative effect of sitagliptin was confirmed by the improvement in the histological structure in H&E stain, the increase of insulin immunoreactivity, the decrease of caspase-3 immunoexpression, and negative iNOS immunoreaction in the pancreatic islets of group 3. These results were in agreement with Ferreira et al. [72], who displayed that sitagliptin treatment improved glycemic state, lipid profile, and inflammation and reduced the histopathological changes induced in both endocrine and exocrine pancreases in ZDF diabetic rat model of type 2 DM.

Different suggestions were postulated for the glucose-lowering effect of GLP-1-based therapies, including DPP-4 inhibitors. Bae [73] declared their role beyond glucose control, including stimulation of insulin secretion and inhibition of glucagon secretion, expansion of the beta-cell mass, and inhibiting beta-cell apoptosis, delay of gastric emptying, and reduction of food intake. In addition, Lee et al. [74] added an anti-inflammatory effect on the islet cells and adipose tissue, contributing to lowering blood glucose levels in diabetes. The immunohistochemical observations of insulin, caspase-3, and iNOS immunostaining, in the present work, seemed to confirm many suggestive mechanisms postulated for the glucose-lowering effect of sitagliptin.

On the other hand, the daily oral administration of metformin in group 4 preserved body weight, decreased serum glucose, insulin resistance, decreased total cholesterol and TGs, as well as islet number and islet size. The ameliorative effect of metformin was confirmed by the histological improvement of the islet morphology in H&E stain, moderate increase of insulin immunoreactivity, and a mild decrease of caspase-3 immunoexpression in the pancreatic islets of group 4. These results were in agreement with a previous study [75], where they used the same HFD/STZ diabetic rat model. The authors revealed that metformin significantly decreased serum glucose concentration, HbA1c, increased plasma insulin level, significantly reduced cholesterol and TGs, the improved microstructure of the pancreatic islets, and increased positive insulin immunoreactivity.

Several mechanisms of the glucose-lowering effect of metformin have been proposed. Previous investigators [76] clarified some mechanistic aspects of metformin, including reduced hepatic glucose output, increased peripheral glucose utilization, decreased fatty acid oxidation, sensitized peripheral tissue to insulin action, increased functional activity of glucose transporters, and increased insulin-mediated insulin receptor tyrosine kinase activity, which then augments a range of insulin signals. Migoya et al. [77] suggested a different mechanism of metformin action mediated through GLP-1. They reported that metformin could stimulate GLP-1 secretion rather than inhibit DPP-4 activity. Furthermore, Liu and Hong [78] reported that metformin could enhance the biological effect of GLP-1 by increasing GLP-1 secretion, suppressing DPP-4 activity, and upregulating the expression of GLP-1 receptor in pancreatic β-cells. Another novel mechanism was postulated by Saisho [79], who reported an anti-inflammatory effect for metformin beyond its glucose-lowering effect.

The findings presented herein supported many suggestions on the mechanistic aspects of metformin. The current results showed increased insulin sensitivity in the metformin-treated group, as indicated by decreased HOMA-IR. The postulated anti-inflammatory effect of metformin was confirmed by attenuation of the iNOS positive immunoreactivity.

The combination of both sitagliptin and metformin in group 5 showed a potent effect on the pancreatic islets more than the independent use of each drug, as confirmed by the biochemical findings, morphometric results of the islet number and islet size, H&E findings, and immunohistochemical observations. It seems that both sitagliptin and metformin have complementary mechanisms of action and additive effects with respect to increasing concentrations of active GLP-1 in plasma, as suggested by Migoya et al. [77]. An extended protective effect of sitagliptin on the liver, an issue that was not addressed in this study, could be considered, which added to metformin effects. In the current animal model of type 2 DM (HFD/STZ), expected liver steatosis should be taken in mind and considered as one of the limitations in future work. Shen et al. [80] reported that sitagliptin reduced insulin resistance in a rat model of HFD/STZ-induced liver steatosis as supported by the suppression of serum TGs and free fatty acids (FFA) and decreased HOMA-IR. Also, the authors reported anti-inflammatory and anti-oxidative effects of sitagliptin via reactivation of the SIRT1/AMPK pathway, which was suppressed by HFD treatment. Suppression of inflammation and oxidative stress elicited in the liver during metabolic diseases, such as DM and obesity, could improve insulin resistance and the altered liver pathology, which added to metformin action.

By comparing and contrasting both monotherapy treatments, there were similarities in maintaining body weight, fasting blood glucose, insulin resistance, and serum TGs. However, sitagliptin was superior in maintaining pancreatic weight and pancreas/body weight ratio, serum insulin, serum cholesterol, islet number and size, insulin immunoexpression in beta cells, and caspase-3 immunoexpression in islet cells. Finally, the different mechanisms of action of each monotherapy encountered in each group of the single treatment modality can be summarized to synergistically promote better glycemic control in the group of combined treatment modality. Also, no evidence of pancreatic injury or pancreatitis was detected on short term treatment of low-dose sitagliptin in the current rat model of type 2 DM, as confirmed by the H&E staining, immunohistochemical observations of iNOS immunostaining, and the biochemical findings of serum lipase.

However, with a low dosage of sitagliptin used herein (10 mg/kg/day), as compared to the literature, caution must be applied as the used dosage is approximately 10-times higher than that used in human (100 mg/day) [81]. Future researches are highly warranted to address such limitation.

## 5. Conclusions

In this animal model of type 2 DM (HFD/STZ rat model), beta-cell damage was mediated via inflammatory and apoptotic pathways, as evidenced by iNOS and caspase-3 immunohistochemistry. Sitagliptin attenuated DM through the preservation of the beta-cell mass and function and decreasing insulin resistance. The current work declared a cytoprotective effect of low-dose sitagliptin on the pancreatic beta cells, which might be due to an improvement of the lipid profile, glycemic state, in addition to its anti-apoptotic and anti-inflammatory effects, as confirmed by the immunohistochemical observations. Furthermore, the obtained data didn’t indicate that sitagliptin, in the administered dose, had a detrimental effect on the pancreatic acini in the current HFD/STZ rat model of type 2 diabetes mellitus. Hence, sitagliptin could be used as an efficient antidiabetic drug during treatment of DM with adequate safety coverage on the exocrine pancreatic tissue, at least when given at a low dose for 4 weeks in rats. However, further studies, using increasing doses of sitagliptin and different methods, are greatly warranted to establish the above-mentioned suggestions.

## Figures and Tables

**Figure 1 biology-09-00006-f001:**
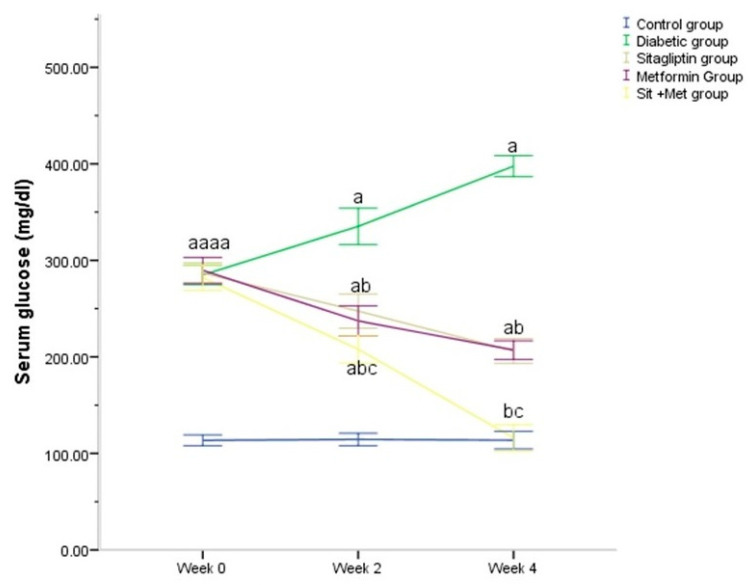
Changes in serum glucose (mg/dL) during the treatment period in the different studied groups. The values were expressed as mean ± SD (n = 10). ^a^: significantly different, compared to the control group. ^b^: significantly different, compared to the untreated diabetic group. ^c^: significantly different, compared to the sitagliptin and metformin-treated groups, using post hoc ANOVA (LSD), *p <* 0.05.

**Figure 2 biology-09-00006-f002:**
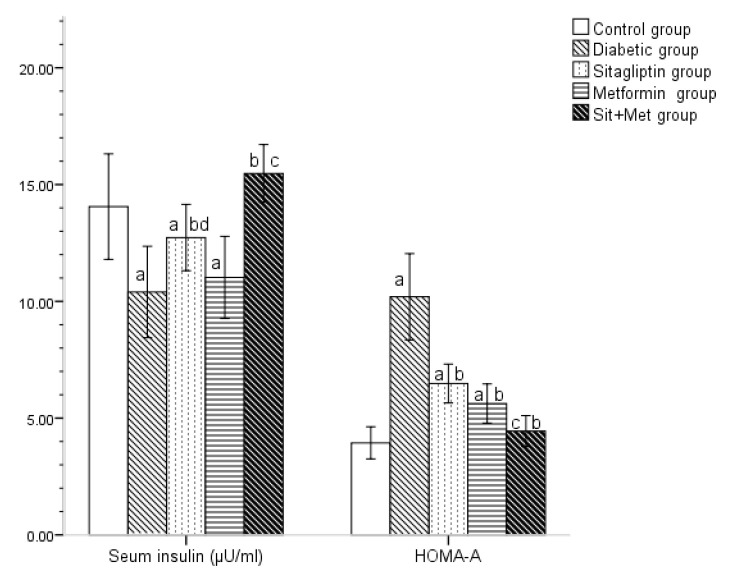
Changes in serum insulin and HOMA-IR in the different studied groups at the end of the treatment period. The values were expressed as mean ± SD (n = 10). ^a^: significantly different, compared to the control group. ^b^: significantly different, compared to the diabetic untreated group. ^c^: significantly different, compared to the sitagliptin and metformin-treated groups. ^d^: significantly different, compared to metformin, using post hoc ANOVA (LSD), *p <* 0.05.

**Figure 3 biology-09-00006-f003:**
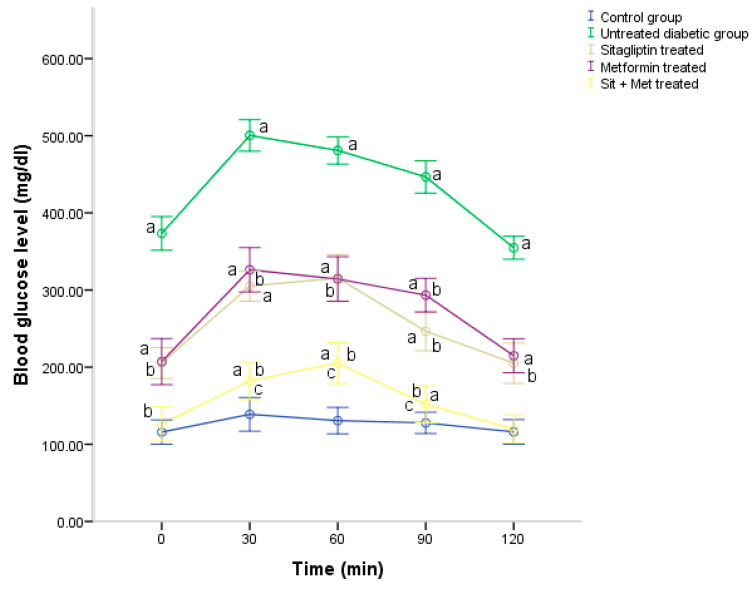
Changes in the blood glucose level (mg/dL) during the conduction of OGTT (oral glucose tolerance test) in the experimental groups at the end of the treatment period. The data were expressed as a mean ± SD (n = 10). ^a^: significantly different, compared to the control group. ^b^: significantly different, compared to the untreated diabetic group. ^c^: significantly different, compared to the sitagliptin and metformin-treated groups, using post hoc ANOVA (LSD), *p <* 0.05.

**Figure 4 biology-09-00006-f004:**
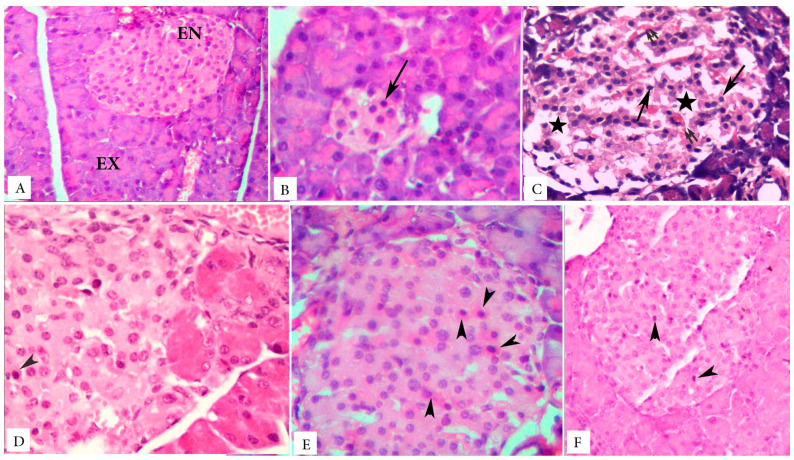
H&E stained microscopic photomicrographs of the different studied groups. (**A**). Control rats show the endocrine pancreas (EN) with histologic features of normal islets. The exocrine pancreatic acini (EX) appear with normal pyramidal lining cells, showing basal basophilia and apical acidophilia. (**B**,**C**). The untreated diabetic group shows shrunken islets with a drastic decrease in the number of their cells. Some islet cells are distorted with vacuolated cytoplasm (stars), small-sized darkly stained nuclei (arrows), and congested blood capillaries (double arrows) in-between. (**D**). The sitagliptin-treated group shows an apparent increase in islet size and normal histoarchitecture. Few islet cells show small, deeply stained nuclei (arrowheads). Pancreatic acini appear closed packed with normal histology (**E**). Metformin-treated group shows an apparent improvement in the islet size, normal cellular arrangement. Apoptotic changes, in the form of small pyknotic nuclei and deeply acidophilic cytoplasm (arrowheads), are seen. (**F**). The combined therapy group shows improved islet size, histoarchitecture, and cytology. Apoptotic nuclei (arrowheads) are also seen. The pancreatic acini appear normal. Magnification: (**A**,**F**) ×200, (**B**–**E**) ×400.

**Figure 5 biology-09-00006-f005:**
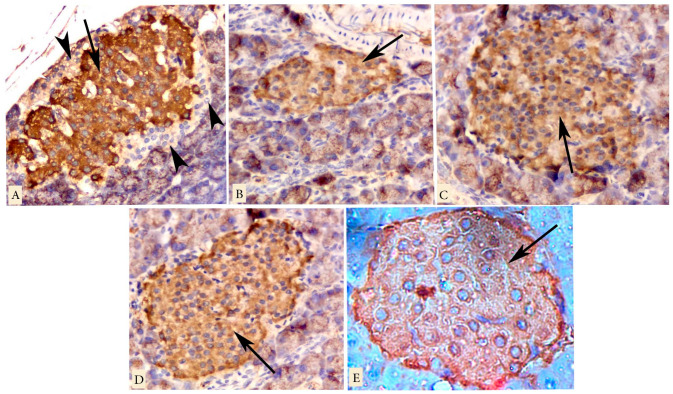
Insulin immunostained photomicrographs of pancreatic islets of the different groups. Arrows mark the brown coloration that is indicative of the positive cytoplasmic reaction. A marked positive reaction is marked in the control group (**A**), in the islet beta cells and surrounded by a peripheral mantle zone of negatively immunostained cells (arrowheads). The shrunken islet, in the untreated diabetic group (**B**), shows a less positive reaction. Sitagliptin-treated (**C**), metformin-treated (**D**), and both-treated (**E**) groups show viable degrees of improvement of insulin immunoreactivity, which is marked in the combined therapy group. Magnification: (**A**–**E**) ×400.

**Figure 6 biology-09-00006-f006:**
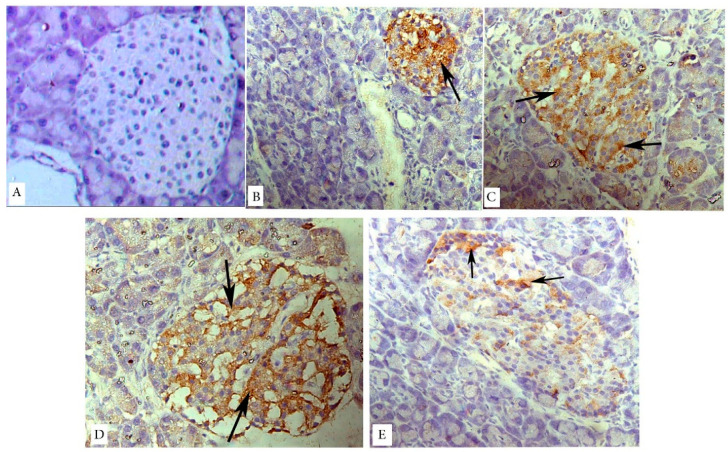
Caspase-3 immunostained photomicrographs of pancreatic islets of the different groups. Arrows mark the brown coloration that is indicative of the positive cytoplasmic reaction. Caspase-3 is not detected in the islet cells of the control rats (**A**). Markedly positive cytoplasmic immunoreaction is noted in the islet cells of the untreated diabetic group (**B**). Treated groups (**C**–**E**) show a decrease in the caspase-3 immunoreaction, with the least reaction in the combined therapy group (**E**). Magnification: (**A**–**E**) ×400.

**Figure 7 biology-09-00006-f007:**
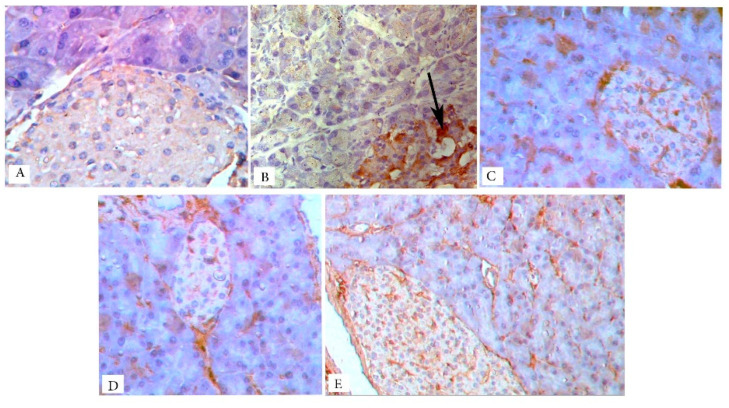
iNOS immunostained photomicrographs of pancreatic sections of the different studied groups. Arrows mark the brown coloration that is indicative of the positive cytoplasmic reaction. The control group (**A**) shows a negative reaction. The iNOS immunoreaction is limited to the islet cells of the untreated diabetic group (**B**), with no reaction detected in the pancreatic acini. All treated groups (**C**–**E**) show negative iNOS immunoreaction in both endocrine and exocrine pancreases. Magnification: (**A**–**D**) ×400 and (**E**) ×200.

**Figure 8 biology-09-00006-f008:**
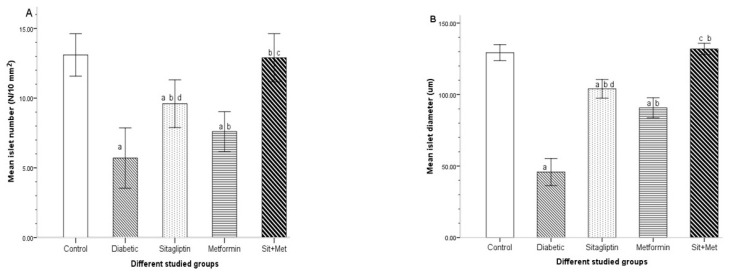
The pancreatic islet number (**A**) and islet size (**B**) in the different studied groups, at the end of the study period. The data were expressed as mean ± SD (n = 10). ^a^: significantly different when compared to control, ^b^: significantly different when compared to diabetic, ^c^: significantly different when compared to sitagliptin and metformin, ^d^: compared to metformin, using post hoc ANOVA (LSD), *p <* 0.05.

**Figure 9 biology-09-00006-f009:**
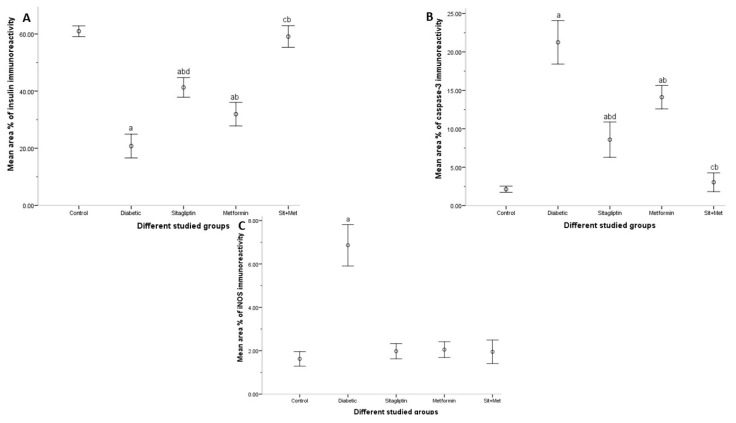
The area percent of insulin (**A**), caspase-3 (**B**), and iNOS (**C**) immunoreactivity in the different studied groups, at the end of the study period. The data were expressed as mean ± SD (n = 10). ^a^: significantly different when compared to control, ^b^: significantly different when compared to diabetic, ^c^: significantly different when compared to sitagliptin and metformin, ^d^: compared to metformin, using post hoc ANOVA (LSD), *p <* 0.05.

**Table 1 biology-09-00006-t001:** Illustration of the workflow of the study.

Stages	Days	Events
**HFD feeding**	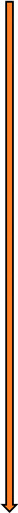	D0-D21	The experimental rats received HFD for 3 consecutive weeks.The rats were weighed every week.FBG was recorded every 3 days.
**STZ injection**	D22	After overnight fasting of the animals, STZ was injected once intraperitoneally at a dose of 40 mg/kg.
**Diabetes mellitus**	D32	Diabetic status of the rats was checked 10 days after STZ injection.Rats with FBG > 270 mg/dL were considered diabetic and were included.
**Initiation of treatment**	D32	All treatment started on the 10th day after STZ injection and continued for 4 consecutive weeks:Sitagliptin: 10 mg/kg/dMetformin: 200 mg/kg/dSitagliptin + metformin: the same dose of both combined
**OGTT**	D60	At the end of the treatment period (4 weeks), an oral glucose tolerance test (OGTT) was done.
**End of experiment**	D60	At the end of the treatment period (4 weeks) and 8 h after performing the OGTT:All animals were lightly anesthetized.Then, cardiac puncture was done to obtain terminal blood samples.The blood sample was stored and then sent for laboratory assay.Finally, the rats were euthanized by cervical decapitation.

**Table 2 biology-09-00006-t002:** Changes in body weight, pancreatic weight, and pancreas/body weight ratio in the different studied groups during the treatment period after the induction of diabetes (ten days after STZ injection). Week 0 (before starting treatment), week 2, and week 4.

Groups	Body Weight (g)			Pancreas Weight (g)	Pancreas to Body Weight Ratio (%)
	Time (weeks)			Time (weeks)	Time (weeks)
	0	2	4	4	4
Normal control	245.15 ± 9.79	261.88 ± 9.62	296.75 ± 9.08	0.800 ± 0.049	0.27 ± 0.014
Diabetic	261.56 ± 8.77 ^a^	305.52 ± 11.33 ^a^	341.80 ± 11.27 ^a^	0.462 ± 0.026 ^a^	0.14 ± 0.008 ^a^
Sitagliptin	262.22 ± 13.68 ^a^	282.25 ± 10.58 ^ab^	318.66 ± 8.97 ^ab^	0.713 ± 0.411 ^abd^	0.22 ± 0.017 ^abd^
Metformin	261.91± 11.08 ^a^	290.51 ± 12.51 ^ab^	318.40 ± 8.65 ^ab^	0.497 ± 0.013 ^ab^	0.16 ± 0.006 ^ab^
Combined	262.74 ± 11.68 ^a^	279.30 ± 16 ^ab^	309.20 ± 9.19 ^abc^	0.802 ± 0.023 ^bc^	0.26 ± 0.009 ^bc^

The values were expressed as mean ± SD (n = 10). ^a^: significantly different when compared to the control group, ^b^: significantly different when compared to an untreated diabetic group, ^c^: significantly different when compared to sitagliptin and metformin, ^d^: significantly different when compared to metformin, using post hoc ANOVA (LSD), *p <* 0.05.

**Table 3 biology-09-00006-t003:** Baseline values of serum insulin and HOMA-IR of the different studied groups.

Groups	Serum Insulin (μU/mL)	HOMA-IR
Normal control	13.91 ± 2.33	3.90 ± 0.71
Diabetic	10.44 ± 2.04 ^a^	7.35 ± 1.49 ^a^
Sitagliptin	10.93 ± 1.86 ^a^	7.73 ± 1.23 ^a^
Metformin	11.07 ± 1.99 ^a^	7.42 ± 1.37 ^a^
Combined	10.28 ± 2.01 ^a^	7.70 ± 1.37 ^a^

The values were expressed as mean ± SD (n = 10). ^a^: significantly different when compared to the control group, using post hoc ANOVA (LSD), *p <* 0.05.

**Table 4 biology-09-00006-t004:** Baseline values (before the beginning of treatment) of the lipid profile parameters and serum lipase in the different studied groups.

Groups	Cholesterol (mg/dL)	Triglycerides (mg/dL)	Lipase (U/L)
Normal control	111.92 ± 5.07	64.25 ± 3.40	514.50 ± 19.80
Diabetic	161.80 ± 8.35 ^a^	138.30 ± 8.63 ^a^	517.55 ± 19.82
Sitagliptin	160.70 ± 6.55 ^a^	137.90 ± 7.48 ^a^	516.70 ± 16.89
Metformin	160.43 ± 7.54 ^a^	137.85 ± 8.33 ^a^	518 ± 19.64
Combined	159.96 ± 8.19 ^a^	137.20 ± 8.35 ^a^	517.30 ± 14.47

The data were expressed as mean ± SD (n = 10). ^a^: significantly different when compared to control, using post hoc ANOVA (LSD), *p <* 0.05.

**Table 5 biology-09-00006-t005:** Changes in lipid profile parameters and serum lipase in the different studied groups at the end of the treatment period (4 weeks).

Groups	Cholesterol (mg/dL)	Triglycerides (mg/dL)	Lipase (U/L)
Normal control	111.31 ± 5.14	64.10 ± 3.41	515.35 ± 19.90
Diabetic	161.35 ± 6.68 ^a^	137.70 ± 8.29 ^a^	518.05 ± 19.54
Sitagliptin	119.10 ± 9.68 ^abc^	71.70 ± 6.44 ^ab^	516.90 ± 16.96
Metformin	128.95 ± 10.12 ^ab^	70.36 ± 5.04 ^ab^	517.95 ± 18.64
Combined	113 ± 7.16 ^bc^	63.67 ± 7.32 ^bd^	517.23 ± 17.36

The data were expressed as mean ± SD (n = 10). ^a^: significantly different when compared to control, ^b^: significantly different when compared to diabetic, ^c^: significantly different when compared to metformin, ^d^: compared to sitagliptin and metformin, using post hoc ANOVA (LSD), *p*-value < 0.05.

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
