# Peer review of "The Biological Impacts of Sitagliptin on the Pancreas of a Rat Model of Type 2 Diabetes Mellitus: Drug Interactions with Metformin"

_biology, 2019, doi:10.3390/biology9010006_

Round 1

Reviewer 1 Report

The manuscript is of interest and addresses a topic of global relevance such as type II diabetes. However, there are aspects that must be improved, such as quality should be improved since the graphs are not properly appreciated.

On the other hand, the results of the rats with CMC are not included in the results section or at least no differentiation is made in untreated diabetic rats.

Reviewer 2 Report

Dear Authors,

This manuscript shows very interesting studies. Overall, I think there are clearly enough new and interesting results for the publication in Biology. However I have some points that must be addressed before paper will be suitable for publication. So, this manuscript should be accepted after minor revision.

Title of Table 1: It is: "illustration of the workflow of the study." It should be: " Illustration of..." Also, the text in the Table 1 is a bit blurred. The quality of this table should be improved. At the end of the line 152 is unnecessary dash. At the end of the lines: 198, 203 there is unnecessary colon. Line 257: It is "Table 4: Baseline values of serum insulin and HOMA-IR of the different studied groups" It should be "Table 4. Baseline values of serum insulin and HOMA-IR of the different studied groups." Also, in line 290 there should be a dot instead of a colon. It is "Figure 2:" It should be "Figure 2." The same in lines 315 and 321. Please correct it and check all manuscript.

Best wishes,

Reviewer

Reviewer 3 Report

In this study, Shawky et al. use a rat model of type 2 diabetes (T2D) to investigate the potential beneficial effects of Sitagliptin, a dipeptidyl peptidase-4 (DPP-4) inhibitor, in preserving pancreatic beta cells, as well as its safety on the exocrine pancreas (given some recent evidence for risk of pancreatitis). The high fat diet/streptozotocin (HFD/STZ) model used has significant limitations in modelling human T2D. Indeed, T2D usually develops in older adults and loss of beta cell mass (induced by STZ) is often seen only in advanced forms of disease. In this study, the rats were young and the loss of beta cell mass seems excessive. Additionally, 4 weeks of treatment with Sitagliptin is too short to call it “chronic”, and lack of observed pancreatitis does not mean immediately good evidence for safety in human.

Unfortunately, the study has many technical issues that greatly limit the ability of correctly interpreting the data. The main points are listed below:

The authors report a very severe reduction in the pancreas weight in the diabetic group (>40%). What is the explanation for this? The HFD itself leads to increase in the body weight (usually also heavier pancreas), and STZ, particularly at the low doze used, should only impact beta cells. This severe reduction in the pancreas size suggests some important toxic effects that greatly reduce the ability of using this as an appropriate model of T2D. The team should have used standard, systematic methods for pancreas sampling. The entire pancreas should have been embedded and sections spaced at 200 micrometers used for analyses. Their method based on collecting 9 fragments can lead to biases that are important in the context of this study. All histology seems very poor quality. There is great amount of background staining everywhere (e.g. insulin stains the acinar cells), the images seem to have different counterstains (for example panel E in figure 4), different levels of zoom-in and cropping and lack scale bars. The non-specific staining observed will definitely reduce the precision of the quantifications provided. The quantification graphs should be placed next to the representative histological images. Beta cell mass is a very important parameter for this study, but is missing. The number of islets and the average size (shown in figure 7), combined with the very severe reduction in the pancreas weight suggest that there is a very important reduction of beta cell mass, well beyond what a typical T2D patient would ever reach. For the OGTT, the first glucose measurement was done at minute 30 after the oral bolus was administered. This is again not a standard way of doing OGTT tests. At least one additional intermediary measurement would be necessary (minute 15). As it is, the OGTT completely misses the peak glucose in the control group (the graph in figure 2 just shows a very flat line). Collecting the samples for biochemical analyses in the serum straight after the finish of the OGTT is not appropriate. The rats should have been left to recover at least 48 hours prior to the terminal collection. Discussion should be shortened and streamlined.

There are also a number of inconsistencies and errors in the text:

The abstract says 50 rats have been used in the study. However the number becomes 40 in the methods. Acronyms should be explained at their first use. See for example GIT mucosa (lane 40). Some words are not spelled correctly, eg. “caspae-3” (lane 166).

Reviewer 4 Report

In this study, the authors have mainly looked at the effect of combined therapy of metformin and sitagliptin on diabetic rats. They conclude that the combined therapy is better. 

The first line of the abstract says - potential adverse effects on pancreas. Then the big question is what was the rationale for using a drug with known adverse effects? This point is justified later in the last paragraph of the introduction. However, the authors should reword the line in the abstract or else justify in the abstract itself. Abstract line 25: I think the authors mean 'sacrificing' and not scarification. The introduction should elaborate a bit on sitagliptin and the studies on it so far. Morever, metformin is not introduced well either. Some background on metformin is required. Also, streptozotocin has not been mentioned in the introduction either. In the introduction, the mechanism of action of metformin is not stated. So it is difficult to understand what the authors exactly mean by synergistic effect of both drugs. Section 2.2 - Please mention the duration of light/dark cycles. After D21, were the rats continued on the high fat diet? It is not clear from the methods and Table 1. For Tables 2 and 3, a graphical representation is better for the reader. Looking at body weights, combined does not seem to be different than sitagliptin or metformin alone. In pancreas weight too, combined is similar to sitagliptin alone. So the conclusion that the combination of both agents is better is untrue. Figure 8 - There should be no lines between the points. That portrays a longitudinal measurement, while these are separate groups. The study is only 4 weeks so the chronic effects of the drugs are not really  being tested. The main concern in the field is the chronic side effects of sitagliptin. So line 452 should clearly state that the current study looked at only acute effects. Also a big caveat is that the authors used only the minimal dosage of sitagliptin, so it is difficult to comment on its safety. Line 473 and 488 - As pointed out earlier, wrt body weight and pancreas weight, there really was no synergistic effect of the drugs. Line 794 - The correct interpretation should be : No evidence of pancreatic injury or pancreatitis was detected in the acute model, with the lowest dose. 

Minor edits:

Line 69 and 72 - The correct word for the past tense of grind is 'ground' and not 'grinded'. Line 78 - No need for 'the', just say 'commercial sources'. Line 198 - The title of section 2.9 should be 'Statistical Analyses'. Analyses is the plural of analysis. Line 603 - spelling error - 'vacuolated'

Round 2

Reviewer 1 Report

The indications made to the manuscript were taken into account.

Author Response

Thanks too much for your valuable comments again.

Reviewer 3 Report

While I understand the limitations of lack of appropriate funding, I don't see any significant improvement in this current version. There are substantial flaws in the experimental set-up, data collection, analysis and interpretation, none of which have been directly addressed by the authors.

Author Response

Please see rebuttal in the attachment.
